# Comprehending the lack of access to maternal and neonatal emergency care: Designing solutions based on a space-time approach

**Núbia Cristina da Silva**[1]**, Thiago Augusto Hernandes Rocha**[1]*****, Pedro Vasconcelos Amaral**[2]**, Cyrus Elahi**[3,4]**, Elaine Thumé**[5]**, Erika Bárbara Abreu Fonseca Thomaz**[6]**, Rejane Christine de Sousa Queiroz**[6]**, João Ricardo Nickenig Vissoci**[1,4,7]**, Catherine Staton**[4,7]**, Luiz Augusto Facchini**[8]

1 Methods, Analytics and Technology for Health (M.A.T.H) Consortium, Belo Horizonte, Minas Gerais, Brazil, 2 Centre for Development and Regional Planning, Federal University of Minas Gerais, Belo Horizonte, Minas Gerais, Brazil, 3 Duke Global Neurosurgery and Neurology Division, Department of Neurosurgery, Duke University School of Medicine, Durham, North Carolina, United States of America, 4 Duke Global Health Institute, Durham, North Carolina, United States of America, 5 Post-Graduate Program in Nursing, Faculty of Nursing, Federal University of Pelotas, Pelotas, Rio Grande do Sul, Brazil, 6 Department of Public Health, Federal University of Maranhão, São Luís, Maranhão, Brazil, 7 Division of Emergency Medicine, Department of Surgery, Duke University School of Medicine, Duke University, Durham, North Carolina, United States of America, 8 Department of Social Medicine, Faculty of Medicine, Federal University of Pelotas, Pelotas, Rio Grande do Sul, Brazil

* rochahernandes3@gmail.com

## Abstract

### Objective

The objective of this study was to better understand how the lack of emergency child and obstetric care can be related to maternal and neonatal mortality levels.

### Methods

We performed spatiotemporal geospatial analyses using data from Brazilian municipalities. An emergency service accessibility index was derived using the two-step floating catchment area (2SFCA) for 951 hospitals. Mortality data from 2000 to 2015 was used to characterize space-time trends. The data was overlapped using a spatial clusters analysis to identify regions with lack of emergency access and high mortality trends.

### Results

From 2000 to 2015 Brazil the overall neonatal mortality rate varied from 11,42 to 11,71 by 1000 live births. The maternal mortality presented a slightly decrease from 2,98 to 2,88 by 100 thousand inhabitants. For neonatal mortality the Northeast and North regions presented the highest percentage of up trending. For maternal mortality the North region exhibited the higher volume of up trending. The accessibility index obtained highlighted large portions of the rural areas of the country without any coverage of obstetric or neonatal beds.

**Data Availability Statement:** The data underlying the results presented in the study are available

here: https://doi.org/10.6084/m9.figshare.12591830.v1.

**Funding:** This research has received funding from the Bill and Melinda Gates Foundation, Grand Challenges Explorations - Brazil: Data Science Approaches to Improve Maternal and Child Health in Brazil Investment (ID OPP1202186 https://www.gatesfoundation.org). Further more funding was given by the Foundation for Research and Scientific and Technological Development of Maranhão - FAPEMA (https://www.fapema.br). National Counsil of the State Foundations for Research Support - CONFAP (#021/2018)/Bill and Melinda Gates Foundation (RCUK-01538/19). Brazil's National Council for Scientific and Technological Development - CNPq (http://www.cnpq.br) granted three fundings: the Grand Challenges Explorations Brasil - Data Science (443834/2018-0); the MS-Decit #14/2016 / CNPq-PQ # 311835/2016-3; and awarded EBAFT with a Research Productivity Scholarship (306592/2018-5 / CNPq N° 09/2018). CA received funding from the Fogarty International Center (K01 TW010000-01A1 https://www.fic.nih.gov/Pages/Default.aspx). The funders had no role in study design, data collection and analysis, decision to publish, or preparation of the manuscript.

**Competing interests:** The authors have declared that no competing interests exist.

## Conclusions

The analyses highlighted regions with problems of mortality and access to maternal and newborn emergency services. This sequence of steps can be applied to other low and medium income countries as health situation analysis tool.

## Significance statement

Low and middle income countries have greater disparities in access to emergency child and obstetric care. There is a lack of approaches capable to support analysis considering a spatiotemporal perspective for emergency care. Studies using Geographic Information System analysis for maternal and child care, are increasing in frequency. This approach can identify emergency child and obstetric care saturated or deprived regions. The sequence of steps designed here can help researchers, and policy makers to better design strategies aiming to improve emergency child and obstetric care.

## Introduction

Despite a global reduction of 44% in maternal deaths from 1990 to 2015, the current global maternal mortality rate of 216 per 100,000 is far from international targets [1, 2]. The maternal mortality gap between high income countries (HICs) and low and middle income countries (LMICs) is still staggering [2].

The challenges to address the maternal and neonatal mortality is different across low (LICs) and middle income countries (MICs). While LICs suffer from limited capacity and resource shortages, the challenge for MICs is partly an issue of coordinating the network of obstetrical services [3]. Currently, MICs account for the greater disparities in access to emergency child and obstetric care (EmCOC) compared to low income countries (LICs) [4]. Challenges in EmCOC coordination have been explored in Brazil, Russia, India, China and South Africa (BRICS) [5–8]. These lack of access problems reflect an unbalanced distribution of health facilities [3]. Geographic access is an important component of maternal and child vulnerability and should be considered when deciding the location and scope of services offered at health care centers [9].

Studies using geospatial analysis for maternal care, comprising techniques capable of integrating location data with maternal outcomes data, are increasing in frequency [10]. When performed at a national level, this approach can assess health care systems and identify EmCOC saturated or deprived regions. A study have found maternal mortality increased in rural settings, non-capital locations within states, and in lower population density locations supporting a need for more services [3]. These macro-level insights can support health policy yet there remains limited studies covering all necessary domains.

The middle income status, unequal deployment of health policy, regional variations, and robust health system level data makes Brazil an ideal country to apply health geographic tools to maternal and neonatal outcomes [9]. From 1990 to 2010 Brazil has seen an overall decrease in maternal mortality, however progress has slowed since 2001 [7]. Child mortality also decreased during the same period. However, there is still a expressive volume of deaths during the neonatal period. At the same time, Brazil refers shows a continued EmCOC access disparities. Reasons for EmCOC limitations include incomplete implementation of policy and a highly unequal society with varying levels of health access [9].

Improved coordination of maternal care and access to EmCOC are imperative to reach the Sustainable Development Goals for maternal and neonatal populations [11]. The objective of this study was to better understand how the lack of access to EmCOC can be related to maternal and neonatal mortality levels. Our study aimed to identify regions facing historic challenges in terms of high mortality trends simultaneously with a lack of access to EmCOC. Our point is that these regions should receive more attention in terms of policies and resource investments. Once the methodological steps performed proof to be able to identify areas to drive investments, these steps could help other countries to perform similar evaluations aiming to improve EmCOC access.

## Methods

### Study design and setting

Our study is ecological with a space time approach. The unit of analysis was the 5565 Brazilian municipalities. We used Brazilian hospital databases to gather evidence regarding the geospatial distribution of maternal and neonatal emergency services. All analyses were done separating neonatal and maternal emergency characteristics. We performed a overlapping geospatial analysis to stratify regions with high levels of neonatal and maternal mortality, simultaneously facing a lack of accessibility regarding EmCOC. Maternal mortality was considered a proxy for lack of prenatal and obstetric care. Neonatal mortality was analyzed as a proxy for lack of neonatal and prenatal care. The mortality trends were defined considering space-time clusters highlighting regions facing historic difficulties with high mortalities rates. The access to EmCOC was mapped using gravity models [12] through the two step floating catchment area (2SFCA) and hotspot approaches. The overlap among regions with high mortality or with a slow mortality decreasing pattern and low EmCOC access would define a group of municipalities that should be prioritized in terms of emergency policies.

### Context

Brazil is the ninth largest economy in the world in 2015 [13]. The country is currently facing a deep recession and has implemented measures to limit investment in the public health system during the next 20 years [14]. Health services in Brazil are mainly provided by a Universal Health System [15]. The country is composed of 26 states and a Federal District, divided between 5 geopolitical regions (Fig 1). A decrease in public health funding increase the need for policies capable of maximizing the impact of scarce resources. Thus, the present work defines a sequence of steps to analyze the health situation concerning EmCOC to create insights capable of supporting policies. This type of need is common in LMICs.

### Data sources and variables

All data analyzed was obtained from public secondary databases, and no approval of Ethics Committee was necessary.

**Mortality data.**   All neonatal and maternal mortality cases were gathered from the Mortality Information System (MIS) [16]. The data analyzed cover a time span ranging from 2000 up to 2015. The rate of neonatal mortality was weighted by the number of born alive extracted from Born Alive Information System [17]. Demographic information about the population size was collected from the Brazilian Institute of Geography and Statistics [18]. The population data was used to estimate the weighted maternal mortality ranged from 2000 to 2015.

**Emergency care data.**   Data of Brazilian hospitals were categorized in two groups regarding neonatal and maternal emergency capabilities. We analyzed data from 925 hospitals of

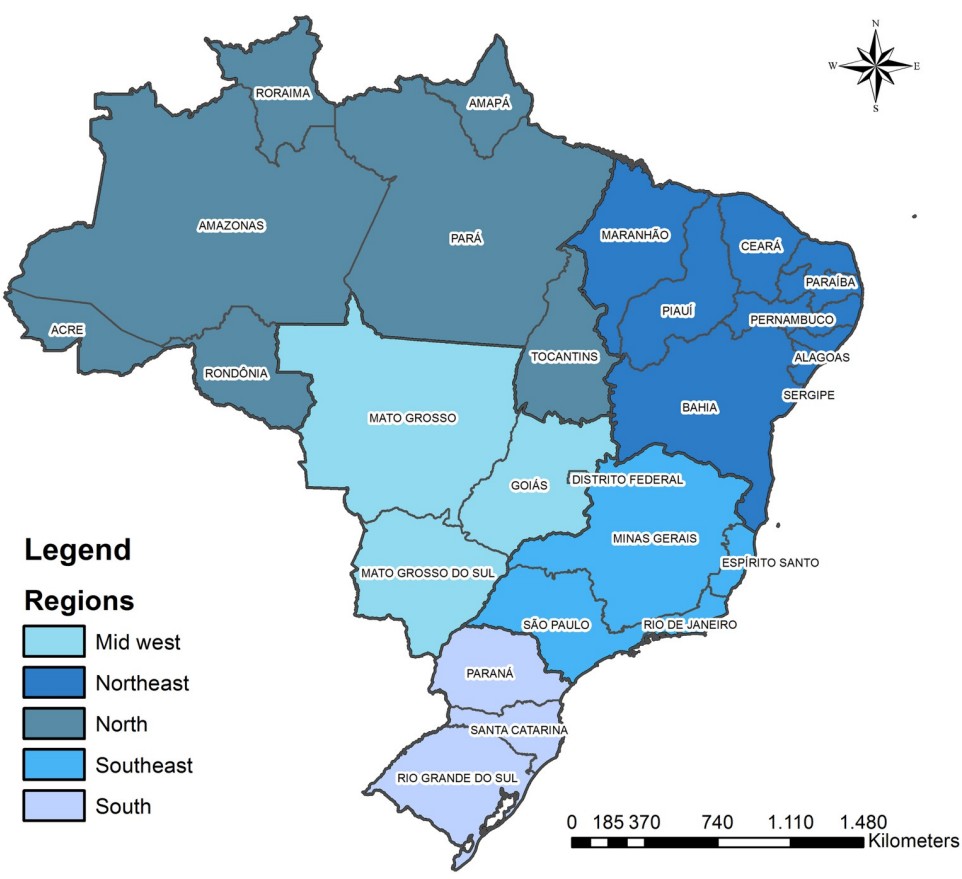

**Fig 1. Brazilian states and regions.**

which 684 offered intensive care unit (ICU) beds dedicated to the neonatal care and 581 offered obstetric beds. All hospitals were geolocated using information provided by the National Registry of Health Facilities at 2015 [19].

## Data analysis

Three analytical steps were performed to characterize the relationship among mortality and accessibility to emergency services. The first step assessed the presence of time dependent clusters of municipalities by levels of neonatal or maternal mortalities. The second step refers to the analysis of accessibility of emergency services regarding hospitals with services dedicated to neonatal and maternal care. The last step comprised an overlap analysis among regions with increasing trends regarding mortality rates or regions with decreasing patterns below the average, concomitantly with low access to emergency services. The municipalities selected with this approach were considered as having more needs in terms of policies concerning EmCOC.

**First step: Space time and hotspot clusters analysis.** All mortality data were geolocated to identify the municipality of residence of the registered death. The maternal deaths were weighted by the population for that given year. The neonatal mortality was weighted by the number of newborns in that given year. Fig 2 details how a space time cube (STC) of mortality rates was created. Each bin in the Fig 2 was analogous to a municipality. The X (latitude) and Y (longitude) axis correspond to the regular spatial distribution of spatial data. Each time slice corresponds to the mortality rates distributed across the Brazilian territory for one year. A

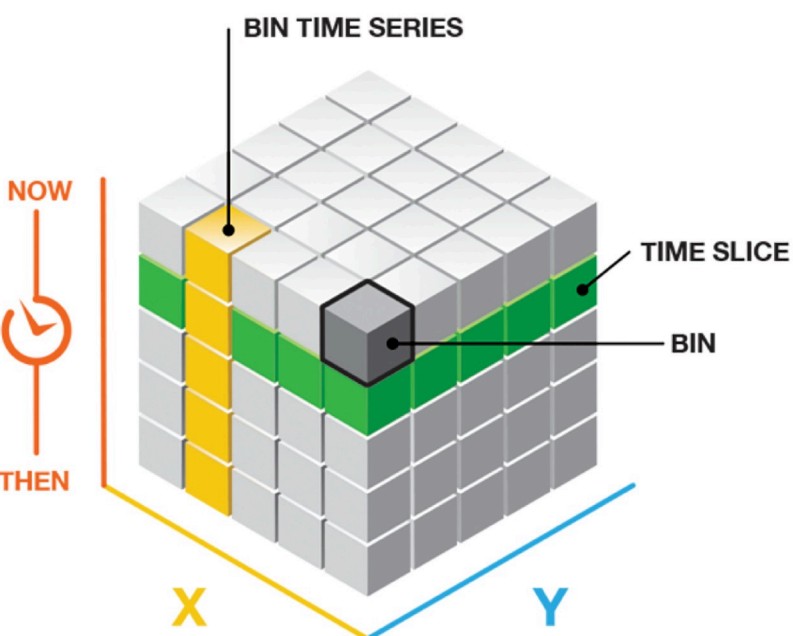

**Fig 2. Structure of the space time cube for a defined location.** Source: Environmental Systems Research Institute [22].

time series describes the evolution of the mortality trends through the years analyzed for each municipality. The outcome of the analyses is an orientation of the time trend tested. Two possible trend orientations were allowed: up or down [20]. The identity the level of statistical significance the Mann-Kendall trend test is performed on every location with data as an independent bin time-series test. The bin value for the first time period is compared to the bin value for the second. If the first is smaller than the second, the result is a +1. If the first is larger than the second, the result is -1. If the two values are tied, the result is zero. A small p-value indicates the trend is statistically significant. The trend for each bin time series is recorded as a z-score and a p-value. The sign associated with the z-score determines if the trend is an increase in bin values (positive z-score) or a decrease in bin values (negative z-score). The STC tool from ARCGIS PRO 10 was used to build the temporal trend regarding neonatal and maternal mortality. Municipalities highlighted as up-trending exhibited a pattern of increased mortality across the years analyzed. The municipalities flagged as down-trending demonstrated the opposite behavior. To define the space time trend, each municipality was compared only with its own data. Green areas highlight an increasing trend regarding the mortality rates in the period. Red areas define a decreasing trend. The outcome categorization of the space time trends for each municipalities were clustered using a Getis-Ord-Gi hotspot analysis [21].

The Gegis-Ord local statistic is given as:

$$G_i^* = \frac{\sum_{j=1}^{n} w_{i,j} x_j = \bar{X} \sum_{j=1}^{n} w_{i,j}}{S \sqrt{\frac{\left[ n \sum_{j=1}^{n} w_{i,j}^2 - \left( \sum_{j=1}^{n} w_{i,j} \right)^2 \right]}{n-1}}} \tag{1}$$

where $x_j$ is the attribute value for feature $j$, $w_{i,j}$ is the spatial weight between feature $i$ and $j$, $n$ is

equal to the total number of features and:

$$\bar{X} = \frac{\sum_{j=1}^{n} x_j}{n} \tag{2}$$

$$S = \sqrt{\frac{\sum_{j=1}^{n} x_j^2}{n} - (\bar{X})^2} \tag{3}$$

The $G_i^*$ statistic is a $z$-score so no further calculations are required.

Red regions (hot spots) highlighted clusters of municipalities with an up-trend considering the indicator analyzed. Additionally a hot spot can also point out to groups of municipalities with a decreasing pattern below the average of the neighboring areas. Blue areas (cold spots) defined groups of municipalities with low trends regarding mortalities rates or with decreasing trend above the neighboring municipalities. Thus, red areas highlight regions with challenges in terms of neonatal and maternal mortality. Whilst blue areas pointed out for municipalities with a better condition in terms of mortality reduction.

**Second step: Current accessibility to emergency services.** We used the 2SFCA approach to assess the accessibility to EmCOC [23, 24]. The 2SFCA create an index of availability of health facilities weighted by population for a specific region. A higher index indicates an increased availability of a specific health service. By using this approach, we were able to assess accessibility to hospitals offering neonatal and maternal care by the interaction of two geographic characteristics: (a) volume of services provided to a determined population and (b) the proximity of services to that population [25]. The coverage area defined as input to the 2SFCA was of 120 kilometers (approximately 2 hours of displacement), corresponding to the recommendations by the Lancet Commission on Global Surgery [26]. A hospital was considered as offering maternal care if there was at least one obstetric bed. For neonatal care, a hospital should have at least one neonatal ICU. A ratio among the selected beds and the population within the distance considered was built as a measure of hospital capacity. A capability index for each hospital was calculated by dividing the average number of beds by the population within the 120 km buffer surrounding the hospital. The equation is:

$$R_j = \frac{S_j}{\Sigma_{k \in \{d_{ij} \leq d_0\}} \frac{D_k}{1000}}$$

where d kj is the distance between k and j, D k is the demand at location k that falls within the catchment, and S j is the availability of beds at location j.

In the second step, the capability indices of all hospitals within 120 km from each municipality's centroid were combined. For each demand location i, we search all supply locations j that are within the threshold distance $d_0$ from location i and sum up the supply-to-demand ratios $R_j$ at those locations to obtain the full accessibility $A_i^F$ at demand location i. The final equation of accessibility index is

$$A_i^F = \sum_{j \in \{d_{ij} \leq d_0\}} R_j = \sum_{j \in \{d_{ij} \leq d_0\}} \left( \frac{S_j}{\Sigma_{k \in \{d_{ij} \leq d_0\}} \frac{D_k}{1000}} \right)$$

This produced an *accessibility index* of EmCOC for *each municipality*. The outcome of the 2SFCA was submitted to Getis-Ord-Gi clusters analysis with FDR correction to categorize

regions according to accessibility levels [21]. Red clusters pointed out areas with a higher access index while blue regions emphasized clusters of low accessibility. The different shades of red and blue highlights the confidence level for each cluster identified.

**Third step: Overlap analysis among mortality hotspots and cold spots of access to emergency services.** The third step corresponded to a spatial overlap of the two previously mentioned analytical steps. The municipalities identified as hotspot clusters regarding the mortalities rates were overlapped with the cold spots by the accessibility index. After these overlapping steps, we had a subset of municipalities categorized as hotspots for mortality rates concomitantly facing scarce access to EmCOC. All analyses were done using the ARCGIS PRO 10.5.

## Results

From 2000 to 2015 Brazil the overall neonatal mortality rate varied from 11,42 to 11,71 by 1000 live births. The Northeast presented an increasing pattern of neonatal mortality for the time-span analyzed. The other regions showed a decreasing volume of deaths through the years considered. The maternal mortality presented a slightly decrease from 2,98 to 2,88 by 100 thousand inhabitants. The distribution of death across Brazilian regions highlighted a higher volume of deaths in Northern and Northeastern regions (S1 Fig).

### Space-time geographical analysis of neonatal and maternal mortality

Fig 3 depicts the STC outcome for neonatal and maternal mortality. For neonatal mortality the Northeast and North regions presented the highest number of municipalities categorized as up trending. For maternal mortality the North region exhibited the higher volume of municipalities classified as up treding (Fig 3A and 3B).

The Getis-Ord-Gi analysis helped to highlight the general clustering trend for the country (Fig 3C and 3D). Several hot spots were pointed out in the North, Midwest, and Southeast regions for maternal mortality. Regarding the neonatal mortality, an expressive hot spot cluster covers an axis from North to South of the country, crossing all Brazilian regions. The hotspots highlighted regions facing challenges to foster the process of mortality reduction for both indicators analyzed, either for being up trend regions or for being downtrend regions with progress below the neighboring average.

### Accessibility to EmCOC

The lack of emergency care services is more prominent for neonatal beds than for maternal care (Fig 4A and 4B). The accessibility index highlighted large portions of the North and Midwest region without any coverage of maternal or neonatal beds. For some regions of the country the hot spot analysis showed opposite patterns of accessibility concerning both indicators monitored. The Northeast region simultaneously exhibited hot spots of availability related to obstetric beds, associated with cold spots regarding neonatal ICUs (Fig 4C and 4D). The South and Southeast regions presented an opposed relation when you compare the accessibility to maternal and neonatal beds.

### Neonatal and maternal mortality trends and lack of emergency services access

For the maternal domain, a group of municipalities in the Southeast, Midwest, and North regions were selected as hotspots, simultaneously with a low accessibility cluster concerning emergency services. For the neonatal care, several groups of municipalities in the North, Northeast, Southeast, Midwest and South regions had low access to neonatal ICU beds, being

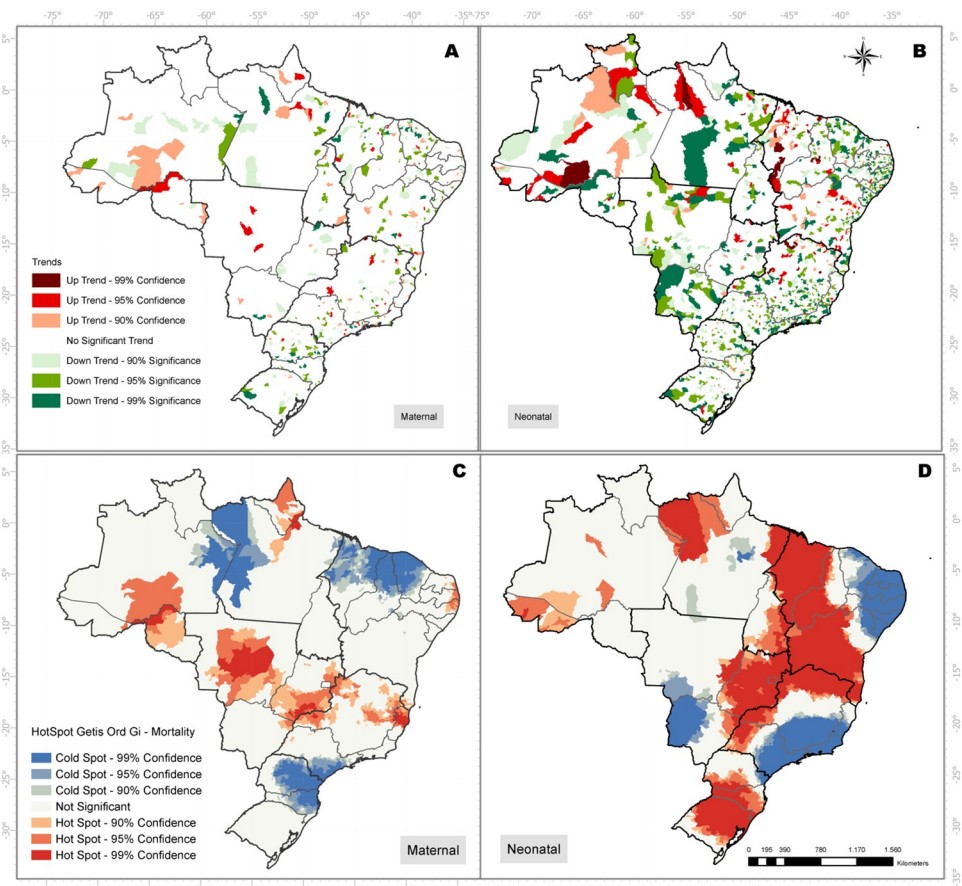

**Fig 3. Space time cluster and hotspot analysis of the mortality rates by type (maternal or neonatal).**

simultaneously categorized as hotspots regarding the neonatal mortality. The states of Maranhão, Bahia, Tocantins and Goiás had a large number of municipalities selected according to this approach (Fig 5).

## Conclusions

The main objective of this work was to better understand how the lack of emergency care regarding EmCOC can be related to maternal and neonatal mortality levels. The use of space time analysis tools conjointly with an accessibility analysis highlighted regions in Brazil facing challenges regarding maternal and neonatal care. To best of our knowledge, the use of geospatial techniques to identify space-time trends in the domain of EmCOC is relatively new with no previous studies reported on PUBMED using this approach conjointly with gravity models of spatial interaction.

The sustainable development goals defined by the United Nations [27] have specific aims dedicated to maternal and newborn health. By 2030, the volume of maternal and newborn preventable deaths should be reduce from all signatory countries. Complications associated with deliveries and newborns are sensitive to access to emergency services [28]. The LMIC setting consists of mixed scenarios captured by the Brazilian reality. The existence of differences and inequities within countries is a common situation. Without effective approaches to perform a health situation analysis, policymakers are unable to make proposals for improvements. The

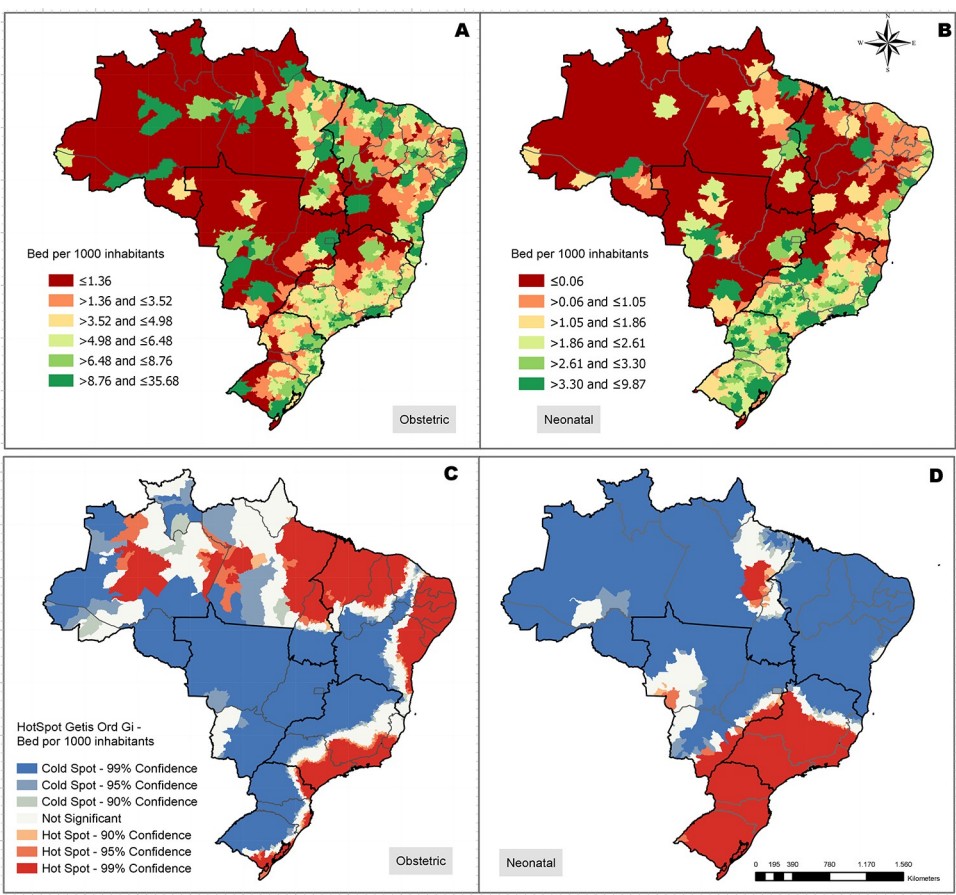

**Fig 4. Accessibility index through 2SFCA and its Getis-Ord-Gi index, hospital beds by type (obstetric or neonatal).**

capacity to quickly identify trends in a dynamic environment and, from these findings, foster evidence based changes are crucial needs to handle EmCOC challenges.

The main obstacle for policymakers aiming to overcome barriers of access related to EmCOC is to get a deep comprehension of the health situation of a specific territory [29]. The

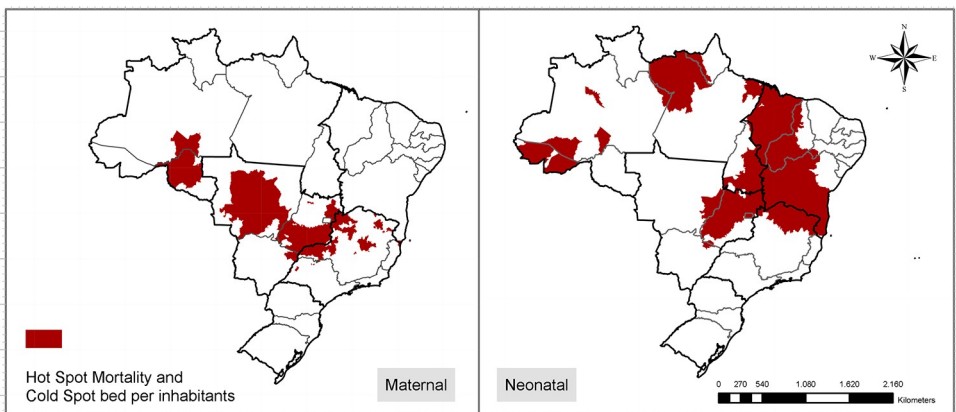

**Fig 5. Municipalities in regions of high mortality and low access to emergency child and obstetric care.**

sequence of steps defined through this paper can be applied to other scenarios with little effort. Additionally, the outcome of the guideline here has the potential to drive the prioritization of emergency health policies to areas facing lack of access and challenges to foster the neonatal and maternal mortality reduction. The availability of financial and human resources to promote improvements regarding EmCOC are usually limited, no matter the country context [26]. Thus the delineation of approaches capable of supporting the process of priority selection can help optimize the use of limited resources to maximize their results. The use of GIS represents a powerful tool aligned with this proposal. Despite its potential, only recently GIS solutions have been applied to address issues concerning maternal and child health [29–31].

The STC analysis demonstrated an apparently dispersed number of municipalities with up trends regarding neonatal and child mortality. Despite this initial perspective, the clusters analysis revealed significant groupings of municipalities facing challenges to reduce the mortality rates considering the 15 years analyzed. The clusters related to maternal mortality defined an axis from the North region to the Southeast portion of the country. The neonatal mortality clusters crossed a vertical axis from North to South. One point demanding our attention was the fact that nearly half of Northeast region was highlighted as an hotspot area. Considering a time span of 15 years, the evidence provided through the STC approach lay emphasis on health problems that must be addressed to guarantee the fulfillment of the sustainable development goal targets. The maternal and neonatal mortalities are complex and multidetermined events. The STC can act only as a screening technique capable of highlighting where undesirable outcomes are happening. This way, more profound studies can be performed to identify the probable causes behind the patterns observed.

Ideally, EmCOC should be available to everyone with complications within 2 hours of travel time to provide lifesaving interventions [28, 32]. The accessibility analysis performed exhibited several areas in the country with no emergency bed within 2 hours of distance. Taking into consideration both indicators analyzed, the existence of obstetric beds is more critical with an axis of low accessibility crossing the Northeast, Southeast and South regions. Thus, the lack of obstetric beds is even larger than highlighted through the cold spot clusters. When considering the access to neonatal ICU, the Northeast and the South regions had several clusters with low availability of beds. The same caution should be taken when analyzing neonatal clusters regarding the inexistence of beds in several areas in the North and Midwest region. The 2SFCA analysis revealed a need to develop policies to address the inequalities in bed distribution across the country.

Access to EmCOC is vital to reduce and manage complications, and prevent adverse maternal and neonatal outcomes [28]. The groups of municipalities highlighted in Fig 5 represent regions characterized as an challenging context in terms of mortality and access to maternal and newborn emergency services. These municipalities presented a historical trend of high levels or slow reduction of mortality. The diagnostics performed throughout this paper can act as starting point to trigger further investigations about the causes behind the situation identified. We observed a relation among low income regions of Brazil and most parts of the municipalities highlighted as having a severe situation. States like Bahia, Maranhão and Minas Gerais had several groups of critical municipalities for both indicators. The outcomes achieved through application of GIS approach can help policy makers to drive additional investigations in these places aiming to improve the understanding of elements explaining the results observed. Another conclusion from the analysis is the need to design different strategies for the country considering the differences observed. Some regions need improvements in the availability of beds while others need more broad approaches to better understand why the temporal trends are pointing to an increase in mortality through time.

Despite the innovative use of space-time trend tool together with a spatial interaction technique the present work has some limitations. We considered the availability of beds as proxies of emergency care and this relation is probabilistic. The mere existence of an emergency bed is no guarantee of appropriate care. Additional elements can help to better characterize the quality of emergency care and should be incorporated in future studies. While the mortality data comprised a time span of 15 years, the information about hospital location relate only to 2015. There was no information available about the geolocation of health facilities for complete time span from 2000 to 2015, and this is a limitation of the Brazilian databases. Despite this, the assessment of the emergency health care network in 2015 still enables policymakers to intervene on the current health facilities structure. Another limitation was the use of euclidean distances to evaluate accessibility. The linear distance is not always similar to the road distances to reach a facility, particularly in the Amazon region. Our idea was to adopt an approach capable to handle the situations observed in large cities, as well as in regions of forest, like in the Amazon. Most academic papers describing methodological developments or improvements such as those for the 2SFCA method underestimate the importance of the geographic details in LMIC as the absence of road access in rural or forest areas, the presence of geographical obstacles like rivers, and other issues present in the Brazilian context. One of the difficulties of calculating spatial accessibility is modeling across vastly different population densities and dispersions. This point is exactly the key strength of the 2SFCA method, once it can be readily applied to both metropolitan and rural areas.

Future studies can go further in the investigation of the causes behind the up trend in mortalities rates for some regions in Brazil. Using additional data about road network, volume of population and others aspects capable to qualify the emergency care will be possible to provide insights and better support the policy making process. The maternal and child mortality are events relatively rare when compared to other types of mortality. Near-miss indicators for maternal and child mortality have been used as proxies to better understand patterns highly associated to deaths. Thus, the use of near-miss indicators for situations may provide a better perspective of the country in terms of lack of care dedicated to EmCOC. Unfortunately was not possible to calculate the near-miss indicators for the present work, but highlight the possibility as future work. Inequity analysis based on income level should also be performed to assess if the high burden of deaths overlaps with poor regions.

The evidence discussed in this work called out for the importance of a GIS approach to produce evidence capable to support the policy making process. Despite its potential solutions, mixing epidemiological, clinical and geographical knowledges are scarce in the literature [29]. The sequence of steps designed here can help governments, health managers and policy makers to better drive interventions aiming to improve EmCOC.

## Supporting information

**S1 Fig. Distribution of mortality rates by region from 2000–2015, Brazil.**
(TIF)

**S2 Fig. Distribution of municipalities by space-time cube trend, by region, from 2000–2015, Brazil.**
(TIF)

## Acknowledgments

The authors would like to acknowledge the Brazilian Ministry of Health for sharing the data.

## Author Contributions

**Conceptualization:** Núbia Cristina da Silva, Thiago Augusto Hernandes Rocha, João Ricardo Nickenig Vissoci.

**Data curation:** Núbia Cristina da Silva, Thiago Augusto Hernandes Rocha.

**Formal analysis:** Núbia Cristina da Silva, Pedro Vasconcelos Amaral.

**Funding acquisition:** Erika Bárbara Abreu Fonseca Thomaz, Rejane Christine de Sousa Queiroz, João Ricardo Nickenig Vissoci, Catherine Staton, Luiz Augusto Facchini.

**Investigation:** Núbia Cristina da Silva, Thiago Augusto Hernandes Rocha.

**Methodology:** Núbia Cristina da Silva, Thiago Augusto Hernandes Rocha.

**Supervision:** João Ricardo Nickenig Vissoci, Catherine Staton, Luiz Augusto Facchini.

**Validation:** Pedro Vasconcelos Amaral, Erika Bárbara Abreu Fonseca Thomaz, Rejane Christine de Sousa Queiroz.

**Visualization:** Núbia Cristina da Silva, Thiago Augusto Hernandes Rocha.

**Writing – original draft:** Núbia Cristina da Silva, Thiago Augusto Hernandes Rocha, Cyrus Elahi.

**Writing – review & editing:** Elaine Thumé, Erika Bárbara Abreu Fonseca Thomaz, Rejane Christine de Sousa Queiroz, João Ricardo Nickenig Vissoci, Catherine Staton, Luiz Augusto Facchini.

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
