## [Decision Letter · Decision Letter 0]

12 Nov 2019

PONE-D-19-18075

Comprehending the lack of access to maternal and neonatal emergency care: Designing solutions based on a space-time approach

PLOS ONE

Dear Dr. Rocha,

Thank you for submitting your manuscript to PLOS ONE. After careful consideration, we feel that it has merit but does not fully meet PLOS ONE’s publication criteria as it currently stands. Therefore, we invite you to submit a revised version of the manuscript that addresses the points raised during the review process.

Additional Editor Comments (if provided):

Please find the comments from the reviewers below. As they highlight, your manuscript addresses a critical topic and your approach is novel and thoughtful. However, there are methodologic and writing concerns which require revision prior to publication.

We would appreciate receiving your revised manuscript by Dec 27 2019 11:59PM. To enhance the reproducibility of your results, we recommend that if applicable you deposit your laboratory protocols in protocols.io, where a protocol can be assigned its own identifier (DOI) such that it can be cited independently in the future. For instructions see: http://journals.plos.org/plosone/s/submission-guidelines#loc-laboratory-protocols

We look forward to receiving your revised manuscript.

Kind regards,

Regan Marsh, MD, MPH

Academic Editor

PLOS ONE

Journal Requirements:

Reviewers' comments:

Reviewer's Responses to Questions

**Comments to the Author**

1. Is the manuscript technically sound, and do the data support the conclusions?

Reviewer #1: Partly

Reviewer #2: Partly

2. Has the statistical analysis been performed appropriately and rigorously? 

Reviewer #1: No

Reviewer #2: Yes

3. Have the authors made all data underlying the findings in their manuscript fully available?

Reviewer #1: No

Reviewer #2: Yes

4. Is the manuscript presented in an intelligible fashion and written in standard English?

Reviewer #1: Yes

Reviewer #2: Yes

5. Review Comments to the Author

Reviewer #1: Núbia Cristina da Silva Rocha et al conduct a very interesting study of an extremely under-represented and important topic: access to specialty hospital beds and the possible association with maternal and neonatal mortality. Furthermore, their use of GIS – a relatively novel methodology – is quite unique in their setting. My main concerns are, while overall a well-designed and incredibly important study, the authors have some serious flaws in the description of their methodology as well as an incomplete/unclear presentation of their results.

Major comments:

1. Incomplete description of their methodology. For example, more details are needed regarding the “geospatial analyses” (page 3, line 66) conducted. The specific tools are listed, but there is no description of specific formulae used or if any specific features/corrections/weighting that may have been used within the tools (i.e. their choice of using 2SFCA and Getis-Ord-Gi vs other related tools or versions of these tools as well as any modifications they may have made for their analyses). Additionally, they do not describe the specific statistical analyses used to determine the level of confidence shown in their figures. Additionally, no mention is made of the completeness/quality of the public databases used as sources for mortality and for resource availability; this is important as this is often a significant limitation in many studies in all settings, particularly lower-resourced settings. Also, it is unclear regarding the “geolocation to each municipality of residence of the death” (page 6, line 111) whether this is where the death actually occurred or if it is the address of the residence of the individual who died. Depending on which it is, suggest this also be listed as a limitation and/or an area for further study as each designation offers very different information and has very different implications post-analysis. Finally, some of the other methods/parameters used or details regarding these methods, such as their use of straight lines for their distances (120km), is listed elsewhere in the manuscript or not listed at all; this type of information should all be listed in the methods section.

2. Incomplete/Unclear presentation of results. For example, in page 6, lines 118-132, the text appears to inconsistently describe what is green/red and up-trending/down-trending/not down-trending at the appropriate rate – i.e. up-trending is described as green here, but red elsewhere. Also, page 6, lines 127-8 describe “municipalities with a decreasing pattern below the average of the neighboring areas”, but this is not fully defined as to how this was determined, and it was not analyzed independently even though it is potentially a different outcome/condition. Also, based on the supplemental information, it appears that a large proportion of the municipalities had no significant change in mortality trends over the fifteen-year period, so it is unclear how this is accounted for/affects the results. Finally, they discuss in their conclusions that a significant amount of the country was not analyzed given the complete lack of beds, but it is unclear why they would not also analyze the mortality in these regions as this would only strengthen their conclusions (page 11, lines 243-5).

Minor comments:

Manuscript: In general, suggest reviewing the grammar and tense throughout the manuscript as it is often inconsistent. A few examples are listed below.

Page 2, lines 44-45 – suggest examining the syntax.

Page 3, line 52 – suggest examining the syntax.

Page 7, line 142 and page 11, lines 239-41 – the “within 120km” recommendation is not included (as far as I can tell) within the source that is listed as the justification for this. Additionally, as the authors point out the use of Euclidian distances does not equal time or actual distance experienced by patients; suggest using consistency in defining this throughout (i.e. not switching between the use of “120km” and “2 hours”).

Page 8, line 168 – mis-spelling.

Page 11, line 245 – mis-spelling.

Page 12, lines 266-276 – agree with the limitations listed here; the authors touch on this, but suggest that the authors discuss even more some of the ways these could motivate/lead to additional studies.

Page 12, line 272 – the authors list the “complete time span” as “2010 to 2015”; but they likely meant “2000 to 2015”.

Conclusions: suggest reviewing this section and making it more succinct and more linear. It is difficult to follow the main conclusion (lack of specialty beds corresponds to increased mortality) and subsequent recommendations, and the authors bring in additional information into this section that is not previously addressed (i.e. relating the results to municipality income, which is not previously addressed or analyzed – page 12, line 257).

Figures:

Figure 1: the legend for this is unclear as the designation for the states and regions is the outline and not the color within each area; suggest changing the legend to have straight colored lines rather than boxes to designate the geographic areas.

Figure 2: the purpose of this figure is unclear as to why this was selected to illustrate a subset of the methodology and other areas of the methodology were not.

Figure 3: suggest explaining a bit more why some of the hot spots do not seem to correlate with the up-trends/down-trends; for example, there seems to be areas of down-trend in the Pará and Mato Grosso states but then these areas are

not highlighted as cold spots.

Figure 4: suggest a review of the legend - should these densities all be greater than or equal (≥) to instead of less than or equal to (≤)? Additionally, the format should have (.) rather than (,). Also, for ease of interpretation should be ranges of whole numbers, rather than discrete numbers with six decimal places? Finally, the inconsistent scale is a bit misleading – the densities listed are approximately 2.7, 4.2, 5.4, 6.9, 9.4, and then 35.7 regarding obstetric beds and 0.8, 1.7, 2.3, 2.9, 3.5, and then 9.9 regarding neonatal beds; suggest revision with a focus on consistency.

Overall, the work is an extremely important topic and a well-written manuscript. Would strongly recommend the authors review and revise the description of their methodology as well as the presentation of their results – specifically to provide additional details of the tools used, to clarify both the description of the results and the figures, and to check for consistency in terminology throughout – and address the minor comments listed above in order to more clearly highlight their main conclusions and to make more robust recommendations for future research and possible interventions. Thank you for this important work.

Reviewer #2: Using the presence of a neonatal ICU and of an obstetric bed as an indicator that emergency obstetric and neonatal service are present seems a bit of a reach, without knowing the staffing, other resources and any other information regarding the facilities. I see this addressed in limitations (line 267-270) by the admission at further study is needed but not sure this is adequate as the study is predicated on this assumption.

If indeed the study is meant only as a “screening tool” to identify areas in need of resources/further intervention, why not just simply at all areas with high maternal/neonatal mortality rates? by limiting "hotspots" to areas with few beds, much of the area with high mortality are eliminated.

“Hotspots” of low access and high mortality rates are nicely defined and highlighted (figure 5) however there is no analysis of the locations that buck this trend (demonstrated in total data in prev figures, areas that have high mortality despite presence of beds, etc). Are the authors hoping to show that, in Brazil, there is a correlation between low access (ie few beds in this case) and high mortality, or is this assumed based on previous knowledge/publications, and the authors are using the “hotspots” to help guide policy, i.e. demonstrate where more resources for maternal and neonatal health should be devoted?

To address the above issues, i would suggest a little more detail showing why the number of beds can be used as a proxy for emergency services in this case, and also some background / reference to any studies that show presence of these beds correlates with improvements in maternal and neonatal mortality. also a clarification of the goal of the paper and what the intention is; the objective states "to better understand how the lack of emergency child and obstetric care can be related to maternal and neonatal mortality levels" but that is not what this study does.

6. PLOS authors have the option to publish the peer review history of their article (what does this mean?). If published, this will include your full peer review and any attached files.

Reviewer #1: No

Reviewer #2: No

---

## [Author Response · Author response to Decision Letter 0]

6 Jan 2020

Dear Dr. Rocha,

Thank you for submitting your manuscript to PLOS ONE. After careful consideration, we feel that it has merit but does not fully meet PLOS ONE’s publication criteria as it currently stands. Therefore, we invite you to submit a revised version of the manuscript that addresses the points raised during the review process.

Additional Editor Comments (if provided):

Please find the comments from the reviewers below. As they highlight, your manuscript addresses a critical topic and your approach is novel and thoughtful. However, there are methodologic and writing concerns which require revision prior to publication.

We would appreciate receiving your revised manuscript by Dec 27 2019 11:59PM. To enhance the reproducibility of your results, we recommend that if applicable you deposit your laboratory protocols in protocols.io, where a protocol can be assigned its own identifier (DOI) such that it can be cited independently in the future. For instructions see: http://journals.plos.org/plosone/s/submission-guidelines#loc-laboratory-protocols

● A rebuttal letter that responds to each point raised by the academic editor and reviewer(s). This letter should be uploaded as separate file and labeled 'Response to Reviewers'.

● A marked-up copy of your manuscript that highlights changes made to the original version. This file should be uploaded as separate file and labeled 'Revised Manuscript with Track Changes'.

● An unmarked version of your revised paper without tracked changes. This file should be uploaded as separate file and labeled 'Manuscript'.

We look forward to receiving your revised manuscript.

Kind regards,

Regan Marsh, MD, MPH

Academic Editor

PLOS ONE

Journal Requirements:

Reviewers' comments:

Reviewer's Responses to Questions

Comments to the Author

1. Is the manuscript technically sound, and do the data support the conclusions?

Reviewer #1: Partly

Reviewer #2: Partly

2. Has the statistical analysis been performed appropriately and rigorously?

Reviewer #1: No

Reviewer #2: Yes

3. Have the authors made all data underlying the findings in their manuscript fully available?

Reviewer #1: No

Reviewer #2: Yes

4. Is the manuscript presented in an intelligible fashion and written in standard English?

Reviewer #1: Yes

Reviewer #2: Yes

5. Review Comments to the Author

Reviewer #1: Núbia Cristina da Silva Rocha et al conduct a very interesting study of an extremely under-represented and important topic: access to specialty hospital beds and the possible association with maternal and neonatal mortality. Furthermore, their use of GIS – a relatively novel methodology – is quite unique in their setting. My main concerns are, while overall a well-designed and incredibly important study, the authors have some serious flaws in the description of their methodology as well as an incomplete/unclear presentation of their results.

We appreciate your comments. We updated the manuscript to better describe our methodological steps, as well as to make the results sections more clear. 

Major comments:

1. Incomplete description of their methodology. For example, more details are needed regarding the “geospatial analyses” (page 3, line 66) conducted. The specific tools are listed, but there is no description of specific formulae used or if any specific features/corrections/weighting that may have been used within the tools (i.e. their choice of using 2SFCA and Getis-Ord-Gi vs other related tools or versions of these tools as well as any modifications they may have made for their analyses). 

We updated the description of the methods used. Additionally, we also added the formula for the 2SFCA, as well as for the space-time cube and the Getis-Ord-Gi clusters analyses. We also described any correction performed, and the analysis parameters selected for each analytical step. 

We acknowledge the reviewer's suggestion. Given the complexities of evaluating access at the macro national level of a country, we understood that the 2SFCA without the enhancements would be conservative to envelop areas without infrastructure, rural areas, and geographical barriers. Many recent ‘improvements’ to the original 2SFCA method have been developed, which generally either account for distance-decay within a catchment or enable the usage of variable catchment sizes. Generally, these improvements aim to address one of two deficiencies of Wang and Luo’s original (crude) 2SFCA method: (1) accounting for distance decay within a catchment; and (2) enabling variable catchment sizes or variable application of distance-decay. Despite some criticisms of the step-decay function having a sudden drop in access at the edge of each zone, these results showed relatively minor differences when comparing the continuous and slow-zone functions, particularly in more sensitive rural areas. A continuous-decay function may intuitively be preferable to a step-decay function, but it is difficult to define an appropriately shaped function that matches ‘real’ behavior of the population (chiefly because of poor empirical evidence for health care seeking behavior). 

Importantly, the application of any distance-decay function creates a strong concentric pattern of high to low access scores out through metropolitan-fringe and into nearby rural areas. This ‘overcorrection’ in metropolitan-fringe areas is a problem when the scope of the analysis considers a large portion of rural areas. That is exactly the situation we faced during the analysis. Our idea was to adopt an approach capable to handle the situations observed in large cities, as well as in regions of forest, like in the Amazon. Most academic papers describing methodological developments or improvements such as those for the 2SFCA method underestimate the importance of the specifics of the geography under consideration - like no road access, presence of geographical obstacles and others present in the Brazilian context. One of the difficulties of calculating spatial accessibility is modeling across vastly different population densities and dispersions. This point is exactly the key strength of the 2SFCA method, once it can be readily applied to both metropolitan and rural areas. 

Additionally, they do not describe the specific statistical analyses used to determine the level of confidence shown in their figures. 

Additional information to describe the statistical tests performed were added. 

Additionally, no mention is made of the completeness/quality of the public databases used as sources for mortality and for resource availability; this is important as this is often a significant limitation in many studies in all settings, particularly lower-resourced settings. 

The details concerning the data limitations were added. The lack of availability of the geolocation form all Brazilian facilities ranging from 2010 to 2015 is an example of a limitation highlighted. 

Also, it is unclear regarding the “geolocation to each municipality of residence of the death” (page 6, line 111) whether this is where the death actually occurred or if it is the address of the residence of the individual who died. Depending on which it is, suggest this also be listed as a limitation and/or an area for further study as each designation offers very different information and has very different implications post-analysis.

We rephrased this period. All deaths are related to the municipality which the person lives. We opt to use the municipality of residence once the lack of access may negatively affect outcomes due to the large distances to be overcome to reach an emergency care facility. 

Finally, some of the other methods/parameters used or details regarding these methods, such as their use of straight lines for their distances (120km), is listed elsewhere in the manuscript or not listed at all; this type of information should all be listed in the methods section.

We revised the manuscript to group this information in the methods section. When not listed we added the details concerning the option adopted. 

2. Incomplete/Unclear presentation of results. For example, in page 6, lines 118-132, the text appears to inconsistently describe what is green/red and up-trending/down-trending/not down-trending at the appropriate rate – i.e. up-trending is described as green here, but red elsewhere. Also, page 6, lines 127-8 describe “municipalities with a decreasing pattern below the average of the neighboring areas”, but this is not fully defined as to how this was determined, and it was not analyzed independently even though it is potentially a different outcome/condition. Also, based on the supplemental information, it appears that a large proportion of the municipalities had no significant change in mortality trends over the fifteen-year period, so it is unclear how this is accounted for/affects the results. 

We updated the text to align the reference of colors and trends. The space time cube analysis highlights the mortality trends between two points of a spectrum - decrease or increase. Brazil has more than 5 thousand municipalities. Identify a trends comparing hundreds of municipalities in a map is an almost impossible task to regular human perception. Thus the cluster analysis intended to depict the general trend emerging from the space time analýsis to define regions where the mortality is increasing or not decreasing as expected due the surrounding conditions observed in a region. The definition of a hot spot clusters using the Getis-ord-Gi approach is based in the average comparisons between neighbors. The Getis-ord-Gi analysis performed with the results of the space time cube helps to identify exactly the regions where the mortality is not decreasing, or is decreasing accordingly to a lower pace than expected, taking into consideration values presented by neighbors municipalities. 

Finally, they discuss in their conclusions that a significant amount of the country was not analyzed given the complete lack of beds, but it is unclear why they would not also analyze the mortality in these regions as this would only strengthen their conclusions (page 11, lines 243-5).

In this specific section we highlighted that a significant portion of two regions were not even analyzed due to its completely lack of beds. The mortality in these municipalities was analyzed.We agree with your suggestion and updated the analysis to take into consideration regions with a complete lack of beds as priority if they overlap with hot spots regions of high mortality. The maps are updated accordingly. 

Minor comments:

Manuscript: In general, suggest reviewing the grammar and tense throughout the manuscript as it is often inconsistent. A few examples are listed below.

Page 2, lines 44-45 – suggest examining the syntax.

We reviewed the text

Page 3, line 52 – suggest examining the syntax.

We reviewed the text

Page 7, line 142 and page 11, lines 239-41 – the “within 120km” recommendation is not included (as far as I can tell) within the source that is listed as the justification for this. Additionally, as the authors point out the use of Euclidian distances does not equal time or actual distance experienced by patients; suggest using consistency in defining this throughout (i.e. not switching between the use of “120km” and “2 hours”).

We reviewed the mentions of euclidean distance and time to reach the health facility to align all descriptions. 

Page 8, line 168 – mis-spelling.

We reviewed the text

Page 11, line 245 – mis-spelling.

We reviewed the text

Page 12, lines 266-276 – agree with the limitations listed here; the authors touch on this, but suggest that the authors discuss even more some of the ways these could motivate/lead to additional studies.

We added a deep discussion concerning the limitations. Additionally we pointed out for other topics not previously comprised in the document. 

Page 12, line 272 – the authors list the “complete time span” as “2010 to 2015”; but they likely meant “2000 to 2015”.

We corrected this phrase. 

Conclusions: suggest reviewing this section and making it more succinct and more linear. It is difficult to follow the main conclusion (lack of specialty beds corresponds to increased mortality) and subsequent recommendations, and the authors bring in additional information into this section that is not previously addressed (i.e. relating the results to municipality income, which is not previously addressed or analyzed – page 12, line 257).

Figures:

Figure 1: the legend for this is unclear as the designation for the states and regions is the outline and not the color within each area; suggest changing the legend to have straight colored lines rather than boxes to designate the geographic areas.

We changed the legend to make it more clear. 

Figure 2: the purpose of this figure is unclear as to why this was selected to illustrate a subset of the methodology and other areas of the methodology were not.

The aim of the figure two is to better represent the logic behind the definition of a space-time cluster, once it combines two dimension of information. We opt to no present the same type of figure for the two step float catchment area once this technique was widely used to asses access to health facilities. On the other hand the space-time cube approach is relatively new in health care assessment. 

Figure 3: suggest explaining a bit more why some of the hot spots do not seem to correlate with the up-trends/down-trends; for example, there seems to be areas of down-trend in the Pará and Mato Grosso states but then these areas are not highlighted as cold spots.

One point that must be discussed is that in Brazil there is a lot of small municipalities. The limits of this municipalities are not highlighted in this figure. For defining a clusters a range of distance is established as limit to search for similar patterns. For the country the average distance defined was of 364km. Thus if a municipality even presenting a similar trend but farway more than 364 km don`t met the necessary criteria to set a cluster. That is the reason why the overlap between the maps in the box A and B doesn`t present a perfect overlap with the hotspot analysis outcome. The use of the cluster analysis (box C and D) is exactly to go beyond the raw pattern and identify statistically significant clusters within the range defined of 364km. 

Figure 4: suggest a review of the legend - should these densities all be greater than or equal (≥) to instead of less than or equal to (≤)? Additionally, the format should have (.) rather than (,). Also, for ease of interpretation should be ranges of whole numbers, rather than discrete numbers with six decimal places? Finally, the inconsistent scale is a bit misleading – the densities listed are approximately 2.7, 4.2, 5.4, 6.9, 9.4, and then 35.7 regarding obstetric beds and 0.8, 1.7, 2.3, 2.9, 3.5, and then 9.9 regarding neonatal beds; suggest revision with a focus on consistency.

We reviewed the legend of the figure accordingly to your suggestions. 

Overall, the work is an extremely important topic and a well-written manuscript. Would strongly recommend the authors review and revise the description of their methodology as well as the presentation of their results – specifically to provide additional details of the tools used, to clarify both the description of the results and the figures, and to check for consistency in terminology throughout – and address the minor comments listed above in order to more clearly highlight their main conclusions and to make more robust recommendations for future research and possible interventions. Thank you for this important work.

We appreciate your contributions. We revised the topics highlighted to improve the way our message are being transmitted. 

Reviewer #2: Using the presence of a neonatal ICU and of an obstetric bed as an indicator that emergency obstetric and neonatal service are present seems a bit of a reach, without knowing the staffing, other resources and any other information regarding the facilities. I see this addressed in limitations (line 267-270) by the admission at further study is needed but not sure this is adequate as the study is predicated on this assumption.

We believe that the process to create evidence capable to support the assessment of access is a incremental approach. The complete lack of hospital beds available by itself defines a situation of care gap. That is the situation for large portions of Brazil. Without solving this point the population living in these regions has no option to get access to EmCOC. Additionally, the regions highlighted in the figure 5 (last figure of the manuscript) are in care gap circumstances during the last decade. We agree that additional points define the adequate care necessary and recommended for all. Despite this, our work was dedicated to assess the first essential step, without which the other topics couldn`t be evaluated. Without hospitals is impossible to discuss if the number of RH staff is enough to provide adequate care. 

If indeed the study is meant only as a “screening tool” to identify areas in need of resources/further intervention, why not just simply at all areas with high maternal/neonatal mortality rates? by limiting "hotspots" to areas with few beds, much of the area with high mortality are eliminated.

Both reviewers of the work raised this question. We adjusted the analysis to incorporate areas with high mortality as well as areas with scarce access to emergency centers. But for this time we not only considered scarce access as the result of the 2SFCA analysis, but also regions with no hospital beds at all. We agreed with the reviewers that the change in the priority area may be suitable to better relate the lack of access with areas presenting a high mortality trend. 

“Hotspots” of low access and high mortality rates are nicely defined and highlighted (figure 5) however there is no analysis of the locations that buck this trend (demonstrated in total data in prev figures, areas that have high mortality despite presence of beds, etc). Are the authors hoping to show that, in Brazil, there is a correlation between low access (ie few beds in this case) and high mortality, or is this assumed based on previous knowledge/publications, and the authors are using the “hotspots” to help guide policy, i.e. demonstrate where more resources for maternal and neonatal health should be devoted?

Our idea aimed to identify regions facing historic challenges in terms of high mortality trends simultaneously with a lack of access to EmCOC. Our point is that these regions should receive more attention in terms of policies and resource investments. Being able to identify areas to drive investments the methodological steps designed could help other countries to perform similar evaluation aiming to better identify regions to be prioritized. 

To address the above issues, i would suggest a little more detail showing why the number of beds can be used as a proxy for emergency services in this case, and also some background / reference to any studies that show presence of these beds correlates with improvements in maternal and neonatal mortality. also a clarification of the goal of the paper and what the intention is; the objective states "to better understand how the lack of emergency child and obstetric care can be related to maternal and neonatal mortality levels" but that is not what this study does.

 We updated the sections highlighted by you. 

6. PLOS authors have the option to publish the peer review history of their article (what does this mean?). If published, this will include your full peer review and any attached files.

Do you want your identity to be public for this peer review? For information about this choice, including consent withdrawal, please see our Privacy Policy.

Reviewer #1: No

Reviewer #2: No

---

## [Editor Report · Decision Letter 1]

26 Jun 2020

Comprehending the lack of access to maternal and neonatal emergency care: Designing solutions based on a space-time approach

PONE-D-19-18075R1

Dear Dr. Thiago Augusto Hernandes Rocha,

We’re pleased to inform you that your manuscript has been judged scientifically suitable for publication and will be formally accepted for publication once it meets all outstanding technical requirements.

Kind regards,

Georg M. Schmölzer

Academic Editor

PLOS ONE
---

## [Editor Report · Acceptance letter]

9 Jul 2020

PONE-D-19-18075R1 

Comprehending the lack of access to maternal and neonatal emergency care: Designing solutions based on a space-time approach 

Dear Dr. Rocha:

I'm pleased to inform you that your manuscript has been deemed suitable for publication in PLOS ONE. Congratulations! Your manuscript is now with our production department. 

Kind regards, 

on behalf of

Dr. Georg M. Schmölzer 

Academic Editor

PLOS ONE